# DSSM: Distributed Streaming Data Sharing Manager [note 1]

**DOI:** 10.3390/s21041344

**Published:** 2021-02-14

**Authors:** Hiroaki Fukuda, Ryota Gunji, Tadahiro Hasegawa, Paul Leger, Ismael Figueroa

**Affiliations:** 1Department of Computer Science and Engineering, Shibaura Institute of Technology, Tokyo 135-8548, Japan; 2Graduate School of Electrical Engineering and Computer Science, Shibaura Institute of Technology, Tokyo 135-8548, Japan; ma19030@shibaura-it.ac.jp; 3Department of Electrical Engineering, Shibaura Institute of Technology, Tokyo 135-8548, Japan; thase@shibaura-it.ac.jp; 4Escuela de Ingeniería, Universidad Católica del Norte, Coquimbo 1780000, Chile; pleger@ucn.cl; 5Escuela de Auditoría, Universidad de Valparaíso, Valparaíso 2340000, Chile; ismael.figueroa@uv.cl

**Keywords:** sensors, robot operating system, streaming data sharing manager, distributed streaming data sharing manager

## Abstract

Developing robot control software systems is difficult because of a wide variety of requirements, including hardware systems and sensors, even though robots are demanding nowadays. Middleware systems, such as Robot Operating System (ROS), are being developed and widely used to tackle this difficulty. Streaming data Sharing Manager (SSM) is one of such middleware systems that allow developers to write and read sensor data with timestamps using a Personal Computer (PC). The timestamp feature is essential for the robot control system because it usually uses multiple sensors with their own measurement cycles, meaning that measured sensor values with different timestamps become useless for the robot control. Using SSM allows developers to use measured sensor values with the same timestamps; however, SSM assumes that only one PC is used. Thereby, if one process consumes CPU resources intensively, other processes cannot finish their assumed deadlines, leading to the unexpected behavior of a robot. This paper proposes an SSM middleware, named Distributed Streaming data Sharing Manager (DSSM), that enables distributing processes on SSM to different PCs. We have developed a prototype of DSSM and confirmed its behavior so far. In addition, we apply DSSM to an existing real SSM based robot control system that autonomously controls an unmanned vehicle robot. We then reveal its advantages and disadvantages via several experiments by measuring resource usages.

## 1. Introduction

Robotics is a hot topic that will support our lives [1,2] and/or cover some tasks that are not suitable for human beings shortly. A robot consists of several sensors and actuators; thereby developing a software system that controls the robot (we call this software by a robot control system) needs a variety of field’s knowledge and experience. Middleware systems such as Robot Operating System (ROS) [3,4] and/or RT-Middleware (RTM) [5,6,7] are created to decrease development costs and are widely used nowadays [3,5]. Streaming data Sharing Manager (SSM) [8] is one of the middleware systems for developing a robot control system using a common Personal Computer (PC). In a robot control system, a single process is in charge of measuring data using a sensor or handling an actuator, meaning that a robot control system consists of multi processes. Suppose that we need to use multiple kinds of sensors and actuators to implement a robot control system. In this robot control system, we should create a process that mainly controls the entire system, which we name the main process. This process needs interprocess communications (shortly IPCs) among other processes that are in charge of measuring sensor data. This is because the main process has to control the entire behavior of the robot referring to the multiple sensor data. Due to data from multiple sensors, data timestamps are important to prevent unexpected system behaviors due to incorrect data processing (we explain the details in Section 2.1).

SSM provides developers useful an API that hides complexities regarding timestamps. A robot control system that uses SSM is also running as multiple processes in a PC, thereby SSM provides shared memories for IPCs among processes. Because of their nature, one process with a heavy load that consumes significant CPU power that might affect other processes, leading to a delay or unexpected control of the robot.

This paper proposes an SSM-based middleware system called Distributed Streaming data Sharing Manager (DSSM), which is implemented in the programming language C++. DSSM uses the existing SSM and enables distributing computing for each process on different PCs without the need to strongly modify existing software systems that use already SSM because of architecture of our proposal. As a result, our proposal ensures that existing software systems that use SSM can run on DSSM because these systems can use SSM directly via DSSM. In addition, DSSM introduces Transmission Control Protocol/Internet Protocol (TCP/IP) communications to divide multiple processes, that was running on a PC, into multiple PCs, then provides a new API that can hide complexities of TCP/IP communications to developers. This paper is a previous work extension [9], where we have developed a prototype of DSSM and shown its behavior. However, it was unclear how using network communications affected the robot control system on limited resources of PCs. Therefore, in this paper, we apply DSSM to an existing real SSM based robot control system that autonomously controls an unmanned vehicle robot that allows us to show its advantages and disadvantages via experiments.

The remainder of this paper is organized as follows. Section 2 describes the SSM architecture and Section 3 explains the DSSM architecture and main components. Section 4 presents an experiment and results that allow us to compare DSSM and SSM efficiencies. Section 5 discusses related work and Section 6 concludes this paper with future works.

**Availability**. The DSSM implementation and the prototype implementation used to carry out the experiment are available on https://github.com/sitRyo/Distributed-Streaming-data-Sharing-Manager.git (revision 8338f5f) (accessed on 13 Feburary 2021).

## 2. Streaming Data Sharing Manager (SSM)

This section describes the background of SSM through its difficulties in controlling autonomous robots. We then briefly explain the SSM architecture.

### 2.1. Difficulties of Controlling Autonomous Robots

An autonomous robot generally uses a variety of sensors that measure several values such as velocities, directions, and distance to objects. A robot control system detects the environment and reacts to prevent unexpected behaviors. Because of requirements of extensibilty that arise from bigger and more complex software, developers have to use several modules to build this of software. For example, suppose a scenario where we need to implement a robot control system that controls an unmanned vehicle robot. The vehicle should stop as soon as it detects a human being in front of it. The robot control system may use a camera to detect a human being, then stop the engine or put on the brake as reactions. In this scenario, we expect to implement this robot control system using, at least, three modules. The first module uses a camera to detect human beings then write the result (e.g., detection or not of the human being(s)). The second module measures and writes the distance to the object that appears in front of the robot using an infrared sensor. Finally, the third module reads these results and reacts (e.g., stop the engine or turn right/left). These modules usually run in different processes. This scenario illustrates, in addition to raw sensor data, that timestamps are relevant. We use Figure 1 to clarify this relevance; two processes (i.e., processA and processB) periodically measure values, in which each interval is different for each sensor (e.g., camera and infrared sensor). Therefore, the main process in Figure 1 needs to choose an adequate pair of values using timestamps: for example, the pair of $s_{2}$ and $t_{3}$ is adequate while the pair of $s_{3}$ and $t_{4}$ is not adequate because of measuring time. Apart from concerning timestamps among multiple processes, we can need to consider timestamps in a single process as well because a process sometimes not only reads the sensor value but also handles it such as image processing, which takes a certain time as Δt. As Figure 2 shows, when we use the handled result (e.g., whether the process detects a human being or not), its timestamp should be *t* not $t^{\prime }$ because the event happened at *t*. As a conclusion, we can say that SSM is a middleware system that hides the complexities of managing timestamps and provides APIs that allow developers to write and read data using timestamps.

### 2.2. SSM Architecture

To illustrate the SSM architecture, we show the usage of SSM in Figure 3. SSM uses shared memories for IPCs because it assumes that all modules are run on a single PC even though each module will be run as different processes. SSM provides APIs, called SSMAPis the summary of which is shown in Table 1, to handle the shared memory: create, read, write, and delete. With these SSMAPis, timestamps are explicitly/implicitly added and used to each measured sensor data. In addition, we call a program that measures sensor data as a sensor handler. Figure 3 shows that “Sensor A”, “Sensor B” and “Sensor C” are sensor handlers, which use SSMAPis inside. In SSM, a sensor handler firstly sends a request to SSM to create a shared memory. The sensor handler also sends an identifier to distinguish the shared memory. When SSM has successfully created the shared memory, it returns the pointer of the shared memory to the corresponding sensor handler. The sensor handler can directly write any data to the shared memory via SSMAPis. In addition, as Figure 3 shows, a program that controls a robot, named User Process, needs to read data stored in shared memories. The program sends a lookup request for finding a certain shared memory to SSM using an identifier, then SSM returns the pointer of corresponding shared memory. Once the user process obtains the pointer, it can directly read data from the shared pointer by specifying timestamps via SSMAPis. The details of complicated processes like creating shared memories are hidden by SSM, implying developers can easily use these kinds of operations. Moreover, SSMAPis are not only be used for creating new shared memory but also be used for other requests to SSM such as terminating or searching existing shared memories. SSM uses a message queue system as an IPC to handle more than one request at a time.

SSM is designed as a middleware system with which developers can implement a robot control system using multiple processes because of several requirements such as customizability and extensibility. At the same time that these requirements are satisfied, SSM minimizes the delay of IPCs through the use of shared memories, making it impossible to run each process on different PCs. As a result, a process that intensively consumes CPU power such as image processing affects the entire behavior of the system because these processes generally use the same CPU. In fact, to illustrate the performance point, we developed a robot control system that controlled an unmanned vehicle robot using SSM through the use of a camera to detect and hit objects; where this camera intensively consumed CPU power. As a result, we could not stop the vehicle before hitting an object even though the camera detected it. To avoid this situation, we decreased the speed of the vehicle, affecting vehicle performance (more in Section 4).

## 3. DSSM: Distributed Data Sharing Manager

This section presents our proposal, DSSM, in detail. We firstly illustrate the DSSM architecture, then explain its main components. We lastly describe the synchronized problem that arises as a side effect introduced by DSSM, and the corresponding solution.

### 3.1. Architecture of DSSM

Figure 4 shows the new architecture that enables to distributing existing processes to different PCs minimizing changes of existing programs. To implement DSSM, we extend the current SSM architecture in which all programs are running as processes. We create an additional IPC using a network to avoid drastically modifying the SSM architecture. In addition, we add the following three components:

**SSMProxy**: This component runs on the PC where SSM is running. An SSMProxy accepts requests from clients (i.e., PConnectors), and it behaves as sensor handlers in existing SSM, meaning that SSM accepts requests from an SSMProxy directly.

**DataCommunicator**: This component is instantiated when a shared memory is created. From a shared memory viewpoint, a DataCommunicartor directly reads/writes data from/to the shared memory.

**PConnector**: This component is used by sensor handlers, which write measured sensor data, to SSM that is running on a different PC. A PConnector directly sends/receives data to/from a SSMProxy using TCP connections. This PConnector provides APIs that are wrappers of SSMAPis and one API for connecting the corresponding DataCommunicator as shown in Table 2.

### 3.2. Behavior of Each Component

We show an application case of DSSM in Figure 5. In DSSM, an SSMProxy, on the same PC where SSM is, runs as an isolated process. When this SSMProxy accepts a request from a PConnector, it creates a child process using a fork system call, then this child process can be able to handle all requests sent from the corresponding PConnector. This is because the original SSM handles requests from more than one sensor handler at a time, then we keep this architecture in our new architecture using the network. An SSMProxy uses SSMAPis to send requests to SSM such as creating or deleting shared memories, meaning that an SSMProxy seems to be a sensor handler from SSM viewpoint.

A sensor handler, which needs to access a shared memory across the network, must use a PConnector in DSSM. The implementation of a sensor handler needs to be changed from SSMAPis to using a PConnector. Even though this change is required, we minimize these changes through a similar interface provided by the current SSMAPis. Because of this, developers may be able to understand PConnectors without a significant additional effort. A PConnector is in charge of communicating with an SSMProxy and a DataCommunicator using a TCP connection. We can also utilize SSMAPis and a PConnector at the same time, meaning that developers can design a software system with more flexibility. For example, PConnectors enable writing processed data to other PCs after writing raw data to a local shared memory using SSMAPis.

When an SSMProxy receives a request to create a shared memory, the SSMProxy sends another request to SSM running on the same PC, then it also creates an instance of a DataCommunicator to handle write/read data from a PConnector. The DataCommunicator will be run as a different thread. This is because a sensor handler in the original SSM can write data to more than one shared memory. A single TCP connection between a PConnector and the corresponding SSMProxy cannot provide this behavior. Therefore, every DataCommunicator opens a socket to make a connection to a PConnector, then writes data sent from the PConnector to the shared memory or sends the data from the shared memory to the PConnector running on different PCs.

With these three additional components, we can run existing software systems that use SSM in a distributed manner minimizing modifications.

### 3.3. Time Synchronize Problem and a Solution

On the one hand, as we mentioned in Section 2.1, controlling autonomous robots requires not only raw sensor data but also their measured timestamps. SSM assumes that all modules of a software system using SSM are running on a single PC. Hence, all modules share the same clock, meaning that there is no concern about time synchronization. On the other hand, in DSSM, these modules are distributed in different PCs in which each PC has its own clock. There are several proposals to address this issue [10,11,12]. As a result, we use Network Time Protocol (NTP) [11] to tackle this issue. We run a NTP Daemon (NTPD) in a PC (e.g., PC1 in Figure 5), then other PCs synchronize the time to this PC. Using two PCs, we measured the offset using NTPD as an average of 140 evaluations of the offset value. Table 3 shows the result where the maximum offset was less than 6 (ms) and the average was 1.41 (ms) as shown with bold and italic style. As we explain in the next section, these results satisfy the requirement of controlling an autonomous robot.

## 4. Experiment and Discussion

Through experiments on an autonomous mobile robot that goes around a map using three Raspberry Pi 3 (RasPi for short), we show the effectiveness of DSSM. First, we describe the hardware used for experiments. Second, we briefly show hardware specifications and explain how to control the unmanned vehicle robot autonomously, and finally, we then describe experiment details and discuss the effectiveness of DSSM.

### 4.1. Hardware Specifications of RasPis and Autonomous Robot

Table 4 presents the specifications of RasPi. In our experiments, we also use one RasPi with SSM to compare both performances. In addition, we show the shape of the unmanned vehicle robot with sensors in Figure 6, where this robot is equipped with two wheels and a battery. Besides, we mainly use two sensors:

**T-frog Motor Driver**: This sensor measures angular velocities of right/left wheels via rotary encoders attached to the wheels.

**Light Detection and Randing (LiDAR)**: This sensor measures the distance to objects that are around the robot.

Table 5 shows the concrete sensors used in the experiments.

### 4.2. Autonomous Robot Control

Using Figure 7, we briefly explain our autonomous robot control. First of all, we need to create a map that has the environment information where the robot moves. Thereby we manually run the robot with LiDAR to collect distances of objects in the environment and calculate the correct path that the robot should run beforehand. We basically execute five processes to autonomously run the robot. Each process exchanges the necessary data using IPCs such as shared memories. Next, we describe these five processes:1DSM is a process that estimates the current position of the robot using distances between the robot and objects in the environment using LiDAR. This process uses the map to calculate and estimate the current position by a scan matching [13]. In this scan matching, DSM tries to match at most 10,000 times in a 10 (ms) cycle, consuming CPU resources intensively. Besides, we apply particle filters for this estimation, meaning that the estimated position depends on probabilities slightly.2Ypspur is also a process that estimates the current position of the robot calculating odometories using angular velocities measured by T-frog Motor Driver. The ypspur process updates the position by adding the last position of the robot and the odometry. This update is enough light compared to the scan matching.3The localizer process finally estimates the position of the robot by combining estimated positions from the processes DSM and ypspur. We use a weighted average for deciding the position which is also used for ypspur to estimate the position in the next round. We use the fixed weight values which might be optimized for more accurate robot control.4Navigator is a process that runs the robot along the path which is defined when we create the map. Navigator makes the decision of what actions should be done such as turn right/left or go straight. This process uses the map and the latest position to decide the action.5Viewer is a process that shows the map and the trajectory of the robot to confirm its behavior. This process also uses the map and the latest position of the robot.

### 4.3. Experiment Details

In the first place, we show the map and predefined path used in the experiments in Figure 8. We set the start and goal points at the same place, then the robot runs the floor in a clockwise fashion. The map seems to be easy for the robot to move along the path because the path consists of straight lines and three times right turns. However, this path is difficult due to a few remarkable objects, where the localizer can help to fix the robot position via DSM. We execute three experiments, where each one runs the robot around the map 10 times using DSSM and SSM:

**Ex 1**: We run the robot with three RasPis ($Pi_{1}$, $Pi_{2}$, and $Pi_{3}$) where the aforementioned processes are running in a distributed manner using DSSM as shown in Figure 9. As mentioned in Section 4.2, the DSM consumes much CPU resource as well as the viewer, therefore they need to be run on dedicated PCs. Consequently, we run the DSM on $Pi_{1}$ and other processes such as the navigator, the ypspur and the localizer on $Pi_{2}$, then the viewer on $Pi_{3}$. In Ex 1, we verify the delay of data transmission with TCP connections, and time synchronization with NTP can be enough small for the autonomous robot control. Besides, we also measure CPU usage to verify how much can DSSM decrease the CPU usage compared to using SSM. For this verification, we run all processes on $Pi_{2}$ with SSM.

**Ex 2**: We add a camera module on $Pi_{3}$ with its driven process, called camp, that receives data from the camera every 10(ms). The camp process writes the result to $Pi_{2}$ using DSSM. In Ex 2, we measure the increase of CPU usage when we add a new sensor and an additional process that will consume much CPU resources. We also add the camera with its camp process to SSM for comparison.

**Ex 3**: We upgrade computers for the robot control from Raspberry Pi 3 to Raspberry Pi 4 to know the effect of computer resources. In Ex 3, all processes are distributed as Ex 2 with DSSM. Table 6 shows the specification of Raspberry Pi 4.

### 4.4. Experiment Results

This section shows the results of our expeirments using trajectories in Figure 10 and measured values in Table 7, Table 8, Table 9, Table 10 and Table 11 and Figure 11, Figure 12 and Figure 13. Firstly, we briefly mention the trajectories of the robot using SSM and DSSM, then explain the details of each experiment.

We show typical trajectories of these experiments in Figure 10. This figure depicts the trajectories with two colors: light gray for DSSM and dark gray for SSM. However, we cannot see the dark gray trajectory because the light gray overwrote the dark gray at a glance, meaning that these trajectories became the same. In other words, the robot control system with DSSM could control the robot autonomously as using SSM.

Looking at each trajectory in experience, we observed differences between DSSM and SSM at corners where the robot had to turn right. We show two figures that depict trajectories at the corner in Figure 11 and Figure 12. These figures depict trajectories with the same colors as Figure 10. In addition, both trajectories also depict the predefined path which is created by connecting way points as the black line. Although Figure 11 shows that trajectories of DSSM and SSM seem to be the same to follow a predefined path, Figure 12 shows that the SSM trajectory got away from the predefined path temporary (e.g., when a wheel might slip) then the robot tried to adjust the position, becoming to follow the predefined path. As a consequence, the robot could follow the predefined path in almost all trials. Next, we detail each experiment:

**Ex 1**: In this experiment, using SSM and DSSM were able to run the robot 9 times from the START to GOAL, meaning that the delay of data transmission and time synchronization could be enough small. In addition, even though the success rate of the robot control is the same (90%) in each case, the use of DSSM could decrease CPU usage as a whole from 50.40% to 35.75% in $Pi_{2}$ as shown with bold and italic style in Table 8. However, iowait using DSSM was increased. This is because DSSM uses additional I/O (i.e., network) and system calls compared to SSM. Moreover, we also observed that the DSM process on SSM consumes approximately 28 (ms) to estimate the current position with scan-matching while DSM on DSSM consumes approximately 38 (ms) on average. This is an unexpected result because DSM on DSSM does not use any network resources. We will discuss this issue in Section 4.5.

**Ex 2**: Similar to the previous experiment, we also tried to run the robot 10 times in each case, respectively. Using DSSM with three RasPis was able to run the robot 8 times from START to GOAL while using SSM was 7 times. We could not observe big differences between using SSM and DSSM from the success rate viewpoint. On the one hand, looking at Table 9, using the camera with SSM increases 10% (from 50.40 to 60.03 as shown with bold and italic style in Table 9) CPU usage compared to that of Ex 1 and the scan-matching in DSM takes approximately 5 (ms) (from 28.74 to 33.68 as shown with bold and italic style in Table 11) more than Ex 1. On the other hand, using the camera with DSSM does not affect the CPU usage of $Pi_{2}$. In addition, STDEV of the scan matching in DSM using SSM becomes double between Ex 1 and Ex 2 while that of using DSSM does not, meaning that the additional process (i.e., camera) also may affect the stability of the entire robot control.

**Ex 3**: Comparing Table 9 and Table 10, using Raspberry Pi 4 decreases CPU usage a bit (2% or 3%) compared to Raspberry Pi 3 while the success rate of the robot control is similar to Ex 2. Furthermore, regarding estimating the position of the robot in DSM, using DSSM consumes more time than that of using SSM, the relation of which is the same as Ex 1 and Ex 2 (SSM < DSSM) shown in Table 11.

### 4.5. Discussions

Through all experiences, the success rate that represents how much the robot runs from START to GOAL was enough high using DSSM (i.e., over 80%), meaning that the offset of NTP for time synchronization and the delay of data transmission with TCP connections are enough small. If the delay exceeds a certain value, the navigator cannot obtain the corresponding data using the delayed timestamp, leading to the fail of robot control. Note that we assumed the delay might be in 1000 (ms) in these expeirments and this assumed value can be changed. Besides, in each case (from Ex 1 to Ex 3), the robot could not run the predefined path completely (100%). This is because the estimation of the position in DSM depends on the particle filter, meaning that the result depends on probabilities slightly [14]. We can adjust the configuration of the particle filter for the map in these experiments; however, we did not do it because the purpose of these experiments did not focus on the logic of autonomous robot control but the applicability of DSSM.

As Figure 13 shows, comparing CPU usages in Ex 1 and Ex 2 using SSM increased approximately 10% CPU usage in $Pi_{2}$ while using DSSM kept the CPU usage in $Pi_{2}$ evenly. This means that, as we expected, the use of DSSM enables distributing the processes that consume CPU resources intensively to different PCs with limited overhead. Although the camp in Ex 2 and Ex 3 just obtained data from the camera and did not use it, the data must be used in real robot control such as object recognition. Therefore, we claim that DSSM should work more effectively if a task needs to strongly use CPU resources. Besides, in Ex 3, DSSM can decrease CPU usage approximately the same as Ex 2 (i.e., 20%), meaning that DSSM will also work effectively when we use PCs with much CPU resource.

Apart from previous discussions, in these experiments, we found unexpected results as we have mentioned in Section 4.4. Even though DSM that estimates the position of the robot uses stored/measured data on the same PC (e.g., $Pi_{1}$), DSM on DSSM took more time than on SSM. This fact is always true through all experiments. Thereby, developers who use DSSM need to consider this delay. More concretely, developers should adjust the size of shared memories to keep old data that might be specified by timestamps measured by DSM.

To sum up this discussion, we can highlight the following contributions of DSSM:1The delays of NTP and data transmission are enough small for robot control.2Using a network slightly affects other processes even though these processes do not directly use the network.3Adding a process that will consume CPU intensively can be run on an isolated PC using DSSM without increasing the CPU usage in the existing PC that controls the robot.

## 5. Related Work

Despite the fact that there is no existing middleware that supports time synchronization in distributed fashion according to the best of our knowledge, we can still discuss proposals close to DSSM.

Robot Operating System (ROS) [3,4] is a middleware system that adopts the publish/subscribe model. ROS processes are represented as nodes, and each node constitutes a graph structure, meaning that each node can directly connect to each other. Once a publisher publishes a topic, all subscribers can receive it. In ROS, a node uses TCP connections for IPCs. Thereby, even though two nodes are running on the same PC, these nodes must connect using TCP connections, leading to a certain delay. In contrast, ROS provides a method to share data in a single process, minimizing the delay. In this case, however, all functions that are required to control a robot should be run in a single process as threads, leading to tightly-coupled relations and dependencies among them, which is not suitable for a modularity viewpoint. Additionately, ROS provides timestamp when a publisher publishes a message, however, this timestamp is the time when the message is sent. In other words, this timestamp is not *t* but $t^{\prime }$ in Figure 2, which is a limitation that might be crucial for several robot controls in real time. Finally, ROS is not in charge of timestamps, therefore developers need to handle them by themselves.

Robot Technology Middleware (RTM) [5,6], following the same line of ROS, is a middleware system, where processes are represented as RT-Components. These processes use CORBA [15] to communicate among them. Since CORBA uses TCP connections, the use of RTM brings a certain delay because of the same reason of ROS. Finally, RTM does not provide abstractions or methods to administer timestamps of measured data as well.

Compared to the previous two middleware systems, as shown in Figure 5, DSSM allows developers to combine two types of IPCs: shared memories and TCP connections. For example, if more than two processes require high-speed communications among them, developers can choose shared memories in a single PC. Instead, if a process will intensively consume resources, developers can easily separate the process on different PCs and choose TCP communications. Moreover, DSSM is in charge of timestamps for measured sensor data. Finally, if developers are already using SSM, these developers can easily introduce DSSM with small changes because our proposal works as an SSM extension.

## 6. Conclusions

With the strong demand for robots nowadays, middleware systems are being developed and used to decrease the development cost of software systems that control robots. SSM is one of such middleware systems that provide a method to write/read data with timestamps. The current SSM assumes that all programs that control a robot run in a single PC as different processes. Thereby one heavy program (process) that intensively consumes CPU resource affects other programs, leading to unexpected results of the robot control.

This paper proposes DSSM that makes it possible to run the programs on different PCs. DSSM provides two types of IPCs: shared memories and TCP connections. Therefore developers can choose and/or combine these IPCs based on the requirements that are new points of DSSM different from existing middleware systems such as ROS. Extending SSM, we propose three additional core components to run programs on different PCs while existing software systems need not be modified as much as possible. We give a prototype implementation of DSSM and apply it to the existing software system that autonomously controls an unmanned vehicle.

We conducted three experiments using Raspberry Pi (3 and 4), and the results of which show the followings:The delays of NTP and data transmission are small enough for appropriate control of the robot.Using network slightly affects other processes even though these processes do not directly use it.Adding a process that will consume CPU intensively can be run on the isolated PC using DSSM without increasing CPU usage in an existing PC that controls the robot.

Considering these findings, users who use DSSM need to consider the implicit side effects of using network even though a process does not use the network directly. Fortunately, these effects can be measured and are generally small; implying these users can modify their systems and/or choose hardware systems considering the effects. Moreover, using DSSM allows users to distribute processes on different PCs easily without much overhead, meaning that these users can add an additional PC to run a CPU intensive process. As a result, DSSM can be applied to any systems that require time sensitive controls such as unmanned drone control.

Regarding future work, we should conduct more experiments with many obstacles for the robot such as real roads or parks where the ground is unstable. Furthermore, although DSSM extends SSM with network, measuring data with timestamp can be implemented on top of ROS/ROS2, meaning that DSSM is not a competitor of ROS/ROS2 but can be a module that uses them. Providing, for example, ros::spinOnce(timestamp) by extending ROS can integrate DSSM features into it.

## Figures and Tables

**Figure 1 sensors-21-01344-f001:**
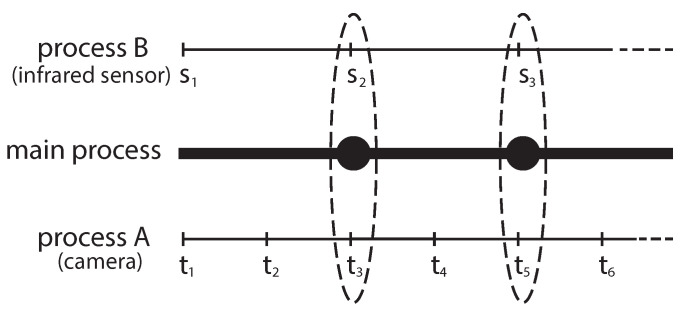
Usage of timestamps.

**Figure 2 sensors-21-01344-f002:**
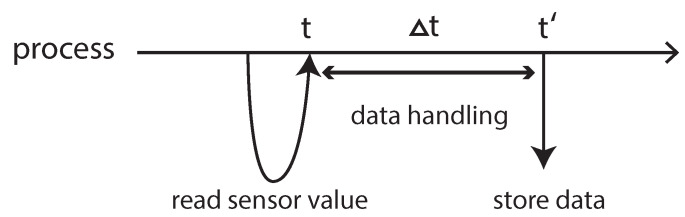
Timestamps in a process.

**Figure 3 sensors-21-01344-f003:**
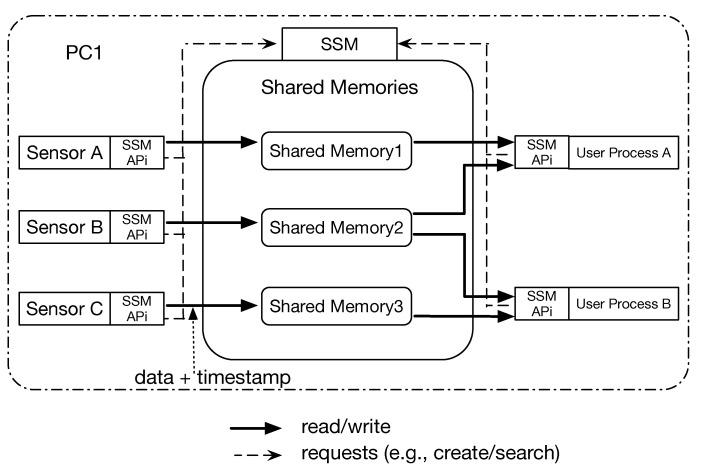
The usage of SSM.

**Figure 4 sensors-21-01344-f004:**
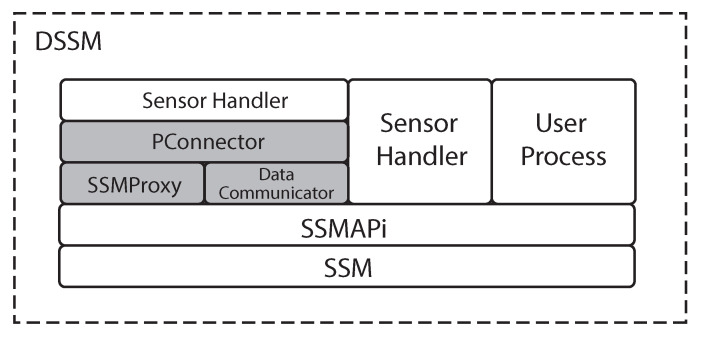
The DSSM Architecture.

**Figure 5 sensors-21-01344-f005:**
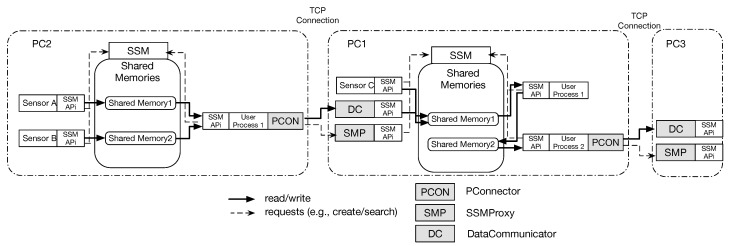
Use case of DSSM.

**Figure 6 sensors-21-01344-f006:**
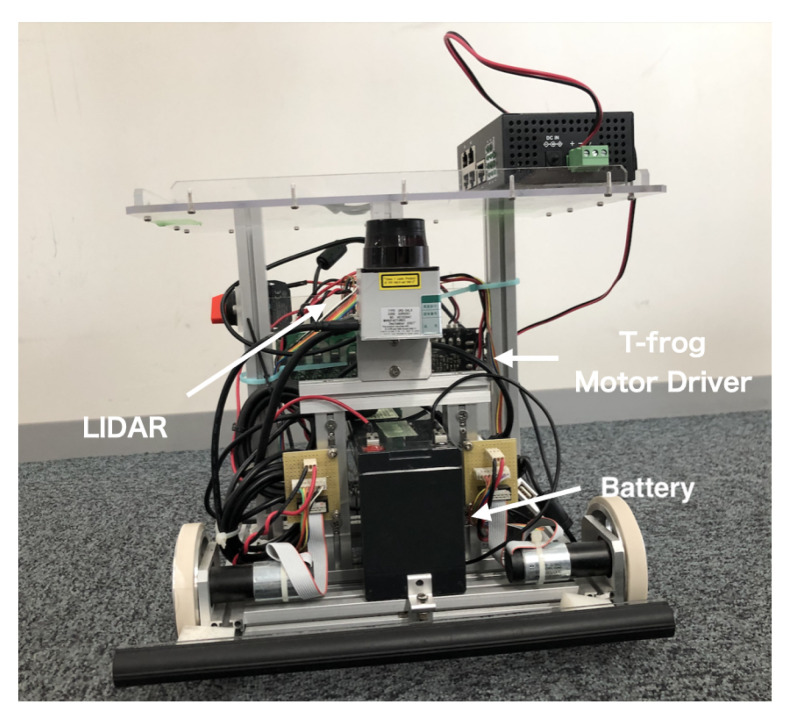
The shape of our unmanned vehicle robot and equipped sensors.

**Figure 7 sensors-21-01344-f007:**
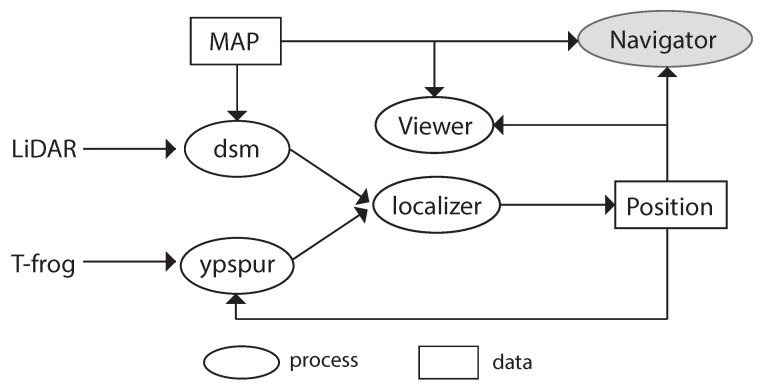
An overview of the autonomous robot control.

**Figure 8 sensors-21-01344-f008:**
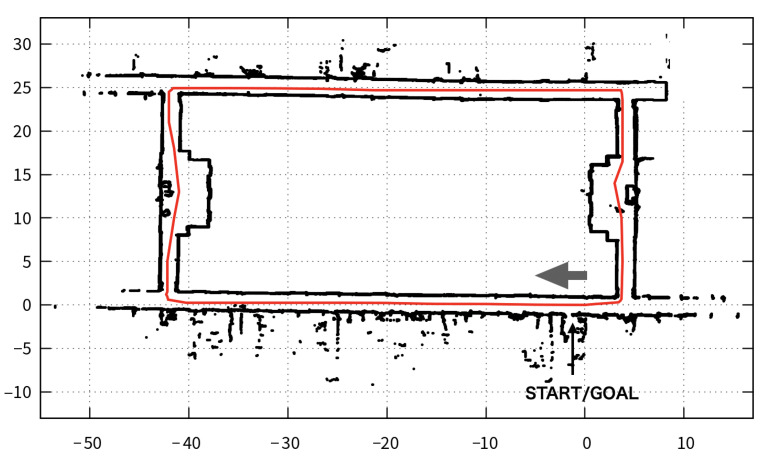
The Map and predefined path for the experiments.

**Figure 9 sensors-21-01344-f009:**
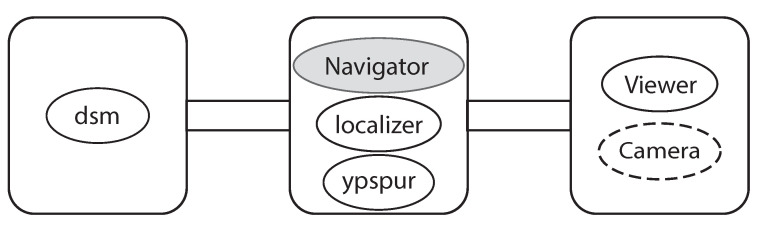
Locations of each process.

**Figure 10 sensors-21-01344-f010:**
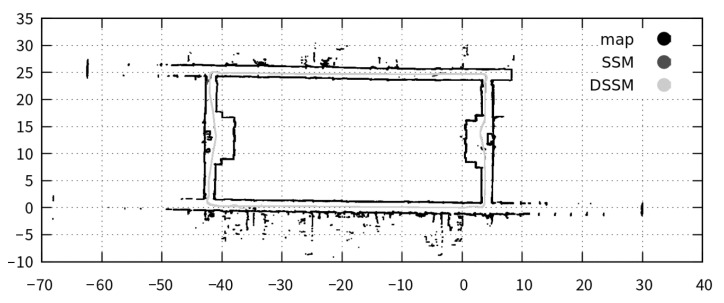
Trajectories of the robot.

**Figure 11 sensors-21-01344-f011:**
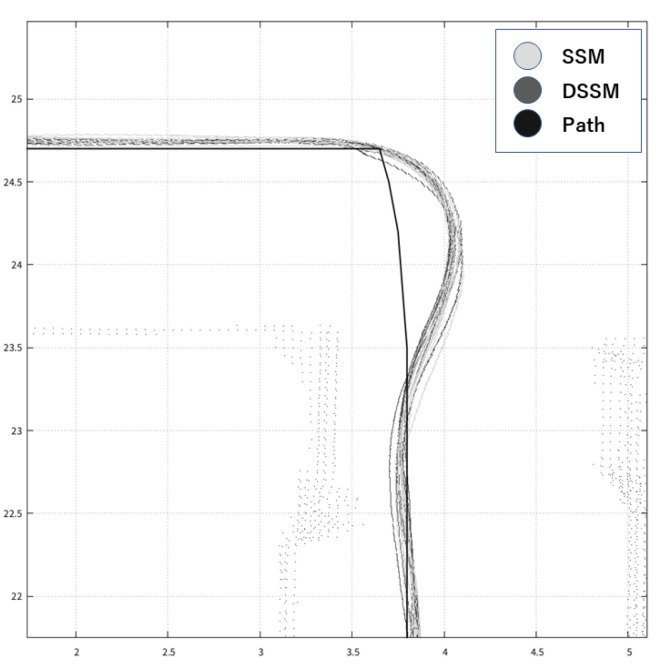
Trajectories following the predefined path.

**Figure 12 sensors-21-01344-f012:**
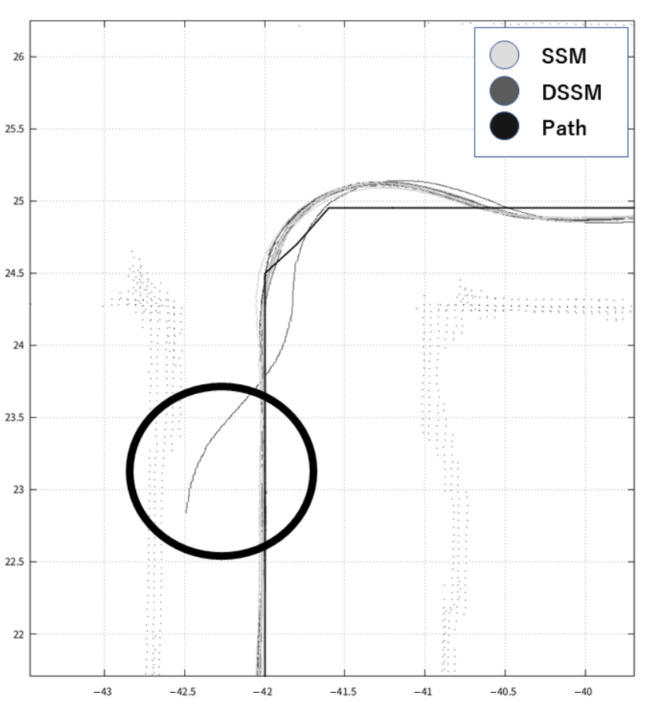
A trajectory recovering the robot position.

**Figure 13 sensors-21-01344-f013:**
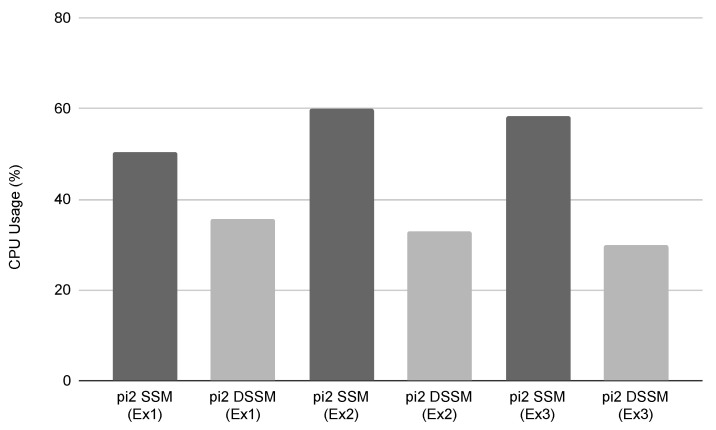
Comparison of CPU usage in Pi2 in all experiments.

**Table 1 sensors-21-01344-t001:** Summary of SSMAPi.

API	Description
write(timestamp)	write data to the shared memory with a timestamp
readLast()	read the last data from the shared memory
readTime(timestamp)	read data corresponding to a timestamp
readNext()	read next data from the last read time
readBack()	read previous data from the last read time

**Table 2 sensors-21-01344-t002:** Summary of APIs provided by PConnector.

API	Description
write(timestamp)	write data to the shared memory with a timestamp
readLast()	read the last data from the shared memory
readTime(timestamp)	read data corresponding to a timestamp
readNext()	read next data from the last read time
readBack()	read previous data from the last read time
createDataCon()	Create and connect to a DataCommunicator

**Table 3 sensors-21-01344-t003:** Time delay between NTPServer and PC using NTPD.

	Average	Max	Min	Standard Deviation
offset (ms)	***1.41***	***5.93***	0.015	1.20

**Table 4 sensors-21-01344-t004:** Specifications of Raspberry Pi 3 model B+.

Feature	Description
Name	Raspberry Pi 3 model B+
CPU	Broadcom BCM2837B0, Cortex-A53 64-bit SoC @ 1.4 GHz
Memory	1 GB LPDDR2 SDRAM
Connectivity	Gigabit Ethernet over USB 2.0 (maximum throughput 300 Mbps)
Storage	8 GB mini SD

**Table 5 sensors-21-01344-t005:** Sensor details.

Sensor	Name
Motor Driver	Tsuji Electronics TF-2MD3-R6
LiDAR	Hokuyo URG-04LX

**Table 6 sensors-21-01344-t006:** Specifications of Raspberry Pi 4 model B.

Feature	Description
Name	Raspberry Pi 4 model B
CPU	Broadcom BCM2711, quad-core Cortex-A72 (ARM v8) 64-bit SoC @ 1.5 GHz
Memory	4 GB LPDDR4-3200 SDRAM
Connectivity	Gigabit Ethernet
Storage	32 GB mini SD

**Table 7 sensors-21-01344-t007:** Success rate of each experiment.

Experiment	PC	Camera	SSM Success Rate (%)	DSSM Success Rate (%)
Ex 1	RaspberryPi 3	No	90	90
Ex 2	RaspberryPi 3	Yes	70	80
Ex 3	RaspberryPi 4	Yes	80	80

**Table 8 sensors-21-01344-t008:** Details of CPU usage in Ex 1.

PC	User	Nice	System	Iowait	Idle	Total
$Pi_{2}$ (SSM)	42.89	0	7.43	0.0083	49.6	***50.40***
$Pi_{1}$ (DSSM)	9.62	0	1.54	0.061	88.90	11.10
$Pi_{2}$ (DSSM)	31.22	0	4.43	0.106	64.25	***35.75***
$Pi_{3}$ (DSSM)	23.15	0	5.95	0.058	70.84	29.16

**Table 9 sensors-21-01344-t009:** Details of CPU usage in Ex 2.

PC	User	Nice	System	Iowait	Idle	Total
$Pi_{2}$ (SSM)	50.18	0	9.73	0.0082	39.7	***60.03***
$Pi_{1}$ (DSSM)	9.49	0	1.56	0.055	88.88	11.12
$Pi_{2}$ (DSSM)	28.66	0	4.24	0.15	66.98	33.01
$Pi_{3}$ (DSSM)	27.39	0	7.40	0.071	65.13	34.87

**Table 10 sensors-21-01344-t010:** Details of CPU usage in Ex 3.

PC	User	Nice	System	Iowait	Idle	Total
$Pi_{2}$ (SSM)	46.83	0	11.31	0.026	41.83	58.17
$Pi_{1}$ (DSSM)	6.98	0	3.81	0.031	89.18	10.82
$Pi_{2}$ (DSSM)	26.26	0	3.64	0.027	70.07	29.93
$Pi_{3}$ (DSSM)	22.18	0	8.86	0.014	68.99	31.01

**Table 11 sensors-21-01344-t011:** Average time of the estimating robot position in DSM.

Experiment	SSM (ms)	STDEV	DSSM	STDEV
Ex 1	***28.74***	8.49	37.55	12.23
Ex 2	***33.68***	15.58	36.58	12.11
Ex 3	14.35	5.39	21.1	5.16

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
