# Peer review of "DSSM: Distributed Streaming Data Sharing Manager†"

_sensors, 2021, doi:10.3390/s21041344_

Round 1

Reviewer 1 Report

The paper is well written and the problem examined is worth of investigation. The solution seems sound.

Since a middleware is proposed I would suggest to better detail the APIs offered by the middleware. Such a description would make the developer aware of the effort to undertake when implementing software components. Moreover, the support provided by the middleware would be much more highlighted.

I suggest to mention the implementation language used for components and the middleware. 

It would be worth comparing, in the related work section, the proposed middleware with other systems or platforms that have provided support to solve the problem of clock synchronisation.

I would suggest to provide a link for accessing the prototype implementation of the middleware.

Reviewer 2 Report

The authors present a distributed streaming data sharing manager (DSSM) for UAV control. This is a distributed version of the streaming data sharing manager (SSM) that runs on a single PC to make how it can be run on different PCs. The authors point out the downsides of SSM while controlling a UAV, when there are many events to keep track of. The delay in processing the events may have negative impact on the performance of the robot. Therefore, the goal here is to distribute the events to be processed to different PCs, through the concepts in concurrent/shared memory parallel computing. The architecture is simple and intuitive. The results are mostly reported for DSSM running in only 2 PCs. This is a severely constrained environment to run the DSSM. This means that it is difficult to evaluate how scalable the DSSM is when the number of PCs increase. Nevertheless, the authors acknowledge this issue and  leave further exploration in this aspect for future work. The experimental results are fine and detailed.

This is an extended version of DSSM architecture and some experiments presented in a conference. It would be nice to have more detailed discussion on the differences with that paper. English editing is recommended to make things clear in different sections. This reviewer found some of the sentences confusing and with grammar issues.    

Reviewer 3 Report

This paper proposes a distributed streaming data-sharing manager that enables distributing processes on SSM to different PCs. The manuscript is based on the article presented at the 2020 IEEE/SICE International Symposium on System Integration, but with the extension of the experimental phase. Please, take into account the following comments and suggestions:

When you start reading the article, the reader gets the feeling that the DSSM is going to be presented as a middleware that offers performance advantages over other systems such as ROS. However, it is not until the end of the article, where it is clarified that DSSM is not a competitor of ROS/ROS2, but can be a module that uses them. Given the large number of people who currently use ROS, an effort should be made to compare the advantages and disadvantages of one system and the other. In addition, it should be explained with more details how they could be complemented or integrated to obtain better performance, considering that ROS enables measuring data with timestamps, as well as the exchange of topics from different PCs though TCP/IP connections.

Section 5 is disjointed with the preceding section and the subsequent section. I recommend integrating part of it in the Introduction (section 1) and another part in the Discussions (subsection 4.5).

In Figure 1, it is not clear if there is only one User Process A, or if they are two different processes.

There seems to be a word missing on lines 162.

There is also a typo on line 172: Ranging instead of Randing.

Round 2

Reviewer 3 Report

The Authors have addressed most of my comments and suggestion. Therefore I recommend the publication of the paper.